# Rare Distant Metastatic Disease of Ovarian and Peritoneal Carcinomatosis: A Review of the Literature

**DOI:** 10.3390/cancers11081044

**Published:** 2019-07-24

**Authors:** Nikolaos Thomakos, Michail Diakosavvas, Nikolaos Machairiotis, Zacharias Fasoulakis, Paul Zarogoulidis, Alexandros Rodolakis

**Affiliations:** 11st Department of Obstetrics and Gynecology, Alexandra Hospital, Gynecologic Oncology Unit, University of Athens, Athens 115 28, Greece; 2Department of Obstetrics & Gynecology, Department of Obstetrics-Gynaecology, Royal Oldham Hospital, Pennine Accute Trust, Oldham OL12JH, UK; 33rd Department of Surgery, “AHEPA” University Hospital, Aristotle University of Thessaloniki, Thessaloniki 546 36, Greece

**Keywords:** Ovarian cancer, distant, metastases, peritoneal cancer

## Abstract

*Background:* Although metastases of ovarian and peritoneal carcinomatosis are most commonly found within the peritoneal cavity, there is a number of other rare distant sites that have been reported. Our goal is to provide an evidence-based summary of the available literature considering the rare distant metastatic sites of ovarian and peritoneal carcinomatosis. *Methods:* A comprehensive search of the literature was conducted, with Medline/PubMed being searched for cases of rare metastatic disease originated from primary ovarian and peritoneal cancer with related articles up to 2019 including terms such as “ovarian cancer”, “metastases”, “peritoneal” and others. *Results:* The most common mechanism of ovarian cancer metastases consists of primarily dissemination within the peritoneal cavity, while, rare and distant sites can either occur at the beginning or during the course of the disease and they are usually associated with hematogenous route and lymphatic invasion, having poor prognosis, with the least common sites being skin, bone, CNS, eye, placenta, central airways, rare lymph nodes, intra-abdominal organs, heart and breast. *Conclusions:* The occurrence of metastatic sites described in this review represents the most common rare distant metastatic sites, and even though their patterns of metastases are still not fully clarified due to the rarity of the reports, they offer valuable information considering the pathophysiology of the disease.

## 1. Introduction

Ovarian cancer is the seventh most common malignancy in women in developed countries, the second most common gynecologic cancer (9.4/100,000), but most importantly, the leading cause of cancer-related deaths in women (5.1/100,000), as the five-year survival rates are below 50%. The mean age of diagnosis is between 55 to 65 years, but it can affect a wide spectrum of ages, with the lifetime risk of developing ovarian cancer being 1.5%. The incidence increases with advanced age and family history of ovarian and breast cancer, which also represent the major risk factors [1,2].

The high mortality rates can be explained by the delayed diagnosis with that vast majority—more than 75%—of affected women being diagnosed at advanced stages of the disease due to lack or non-specific symptoms, and only 15% is confined to the primary site at the time of diagnosis. The most important prognostic factor, besides the histologic type, is the stage of cancer, as more than 60% has already metastasized to other tissues at the time of diagnosis [3].

Ovarian carcinoma usually remains locoregionally confined and metastasizes by direct extension from the ovarian tumor to neighbor pelvic organs, such as the bladder (17%) and colon, or by detachment of cancer cells that spread via transperitoneal dissemination. Exfoliated cells are transported by peritoneal fluid and can metastasize in every intraperitoneal structure, such as the peritoneum and omentum (86%), bowel (50%), and spleen (20%). Metastases through the vasculature are less common in ovarian cancer (16%), with the most common sites being pleura (33%), liver (26%), and lung (15%) [4]. On the contrary lymphatic metastases both in ovarian and peritoneal cancer, is quite common as 70% of ovarian cancer patients present with pelvic and/or para-aortic lymph node involvement (ovarian and peritoneal carcinomas are indistinguishable and are considered a single clinical entity with shared pathogenesis and clinical features) [2,3,5].

Although, as previously mentioned, metastases of ovarian and peritoneal carcinomatosis are most commonly found within the peritoneal cavity, there is a number of other rare distant sites that have been reported. These can either occur at the beginning or during the course of the disease and they are usually associated with hematogenous route and lymphatic invasion, having poor prognosis, with the least common sites being mainly skin, bone, central nervous system (CNS), eye, breast, bronchus-trachea, heart-pericardium, rare lymph nodes and extremely rare intra-abdominal sites (Figure 1, Table 1) [5,6,7,8].

## 2. Methods and Objectives

A comprehensive search of the literature was conducted using the following keywords: “ovarian cancer”, “distant metastases”, “peritoneal” and others. Medline/PubMed was carried out for our primary search.

The aim of this review is to provide an evidence-based summary of the available literature considering the rare distant metastatic sites of ovarian and peritoneal carcinomatosis and their early recognition based on the fact that further update of current bibliography may expand our knowledge and be essential in the improvement of the diagnostic and therapeutic approach. 

## 3. Pathways and Metastatic Sites of Ovarian and Peritoneal Cancer

### 3.1. Pathophysiology of Ovarian and Peritoneal Cancer Metastases

The exact etiology for ovarian-peritoneal cancer is still not fully exploited, and because such cancer types tend to be diagnosed at advanced stages, the early molecular events underlying development are not fully clarified. A series of studies suggest that certain factors, including genetic predisposition, nulliparity, and benign inflammatory diseases, are involved in ovarian cancer formation. Recent data reveal that the presence of different rare metastatic pathways of ovarian cancers depends on the different histologic types, especially the four major representative histologic types that include serous, mucinous, endometrioid, and clear cell types, while transitional cell and squamous cell cancers are reported at low frequency [3,22,55].

Until recently, the common assumption proposed that ovarian tumors were developed via altered epithelial cancer cells arising from the surface of the ovary. More recently, the "incessant ovulation hypothesis" suggests that during ovulation, rupture and repairing of the ovarian surface epithelium lead to continuous genetic alterations that are able to develop tumors. Then, the formed tumor cells loose contact and exfoliate into the peritoneal cavity where they float in the peritoneal fluid and are carried all over the peritoneal cavity attaching to the adjacent peritoneal organs, eventually forming metastatic tumors [56,57].

Another model of ovarian cancer development proposes that circulating gonadotrophins and other hormones (estrogens/androgens), promote the ovarian epithelial cells to develop cancerous formations [58].

Moreover, based on latest findings, there has been a progressive shift towards the theory of the fallopian tube as the origin of high grade serous ovarian carcinoma (HGSC) [59]. HGSCs express proteins typical of secretory cells of the fallopian tube (PAX8) and frequently, they contain mutations in the tumor suppressor gene TP53. Based on that and upon evidence emerging after histologic evaluation of salpingo-oophorectomy specimens in women with breast cancer (BRCA) gene mutations, data indicate that there is a continuum of morphologic tubal alterations and dysplastic lesions—serous tubal intraepithelial carcinoma (STIC)—and HGSCs, which lead to ovarian cancer formation. Thus, STICs may be characterized as the prototypical precursor of such ovarian carcinomas. These results suggest that ovarian cancer is, in fact, a disease of the fallopian tubes, and the development of p53 signatures and STICs constitute the early events of carcinogenesis, approximately five to seven years before the progression to HGSCs [60,61,62,63].

On the contrary, peritoneal carcinogenesis has three major hypothetic models considering the molecular mechanisms of metastases and especially for mesothelioma, the main representative of primary peritoneal cancer. The “oxidative stress theory” is based on the fact that the mast cells that engulf asbestos fibers produce large amounts of free radicals due to being unable to digest the fibers, and epidemiological studies supporting that iron-containing asbestos fibers appear more carcinogenic. Moreover, the “chromosome tangling theory” indicates that asbestos fibers damage chromosomes during cell division and the “theory of adsorption of many specific proteins and carcinogenic molecules” in which asbestos fibers in vivo concentrate proteins and/or chemicals such as the components of cigarette smoke [64,65,66]. 

All these factors mentioned are proven to play a major role in both the physiologic function of the healthy ovary and that of ovarian and peritoneal cancer cells. 

### 3.2. Rare Metastatic Sites

#### 3.2.1. Central Nervous System

Although brain metastases are a common site and complication in lung and breast cancer, it is considered a rare event in ovarian and peritoneal cancer, the incidence of which has been reported to be 0.3–12% among a number of studies. CNS metastases mainly occur via hematogenous and lymphatic spread with reports of direct invasion of the nervous tissue after bone involvement [67]. It is not related to the histology or grading of the ovarian tumor and reveals poor prognosis. Moreover, the latest data suggest an increasing number of patients with brain metastases, especially from primary epithelial ovarian cancer [9,10,11,12]. This could reflect the higher efficiency of imaging technology, but also it could be the result of longer survival rates in such patients. In 31 case series including 460 patients, CNS metastases occurred between 11 to 46 months, with median time after diagnosis being 24.3 months [11]. However, proven CNS metastases developing in more than ten years after the diagnosis of ovarian cancer have also been reported [68]. Porzio et al. reported a review of five studies where brain metastases occurred in 4% of the total patients (712) who were diagnosed with epithelial ovarian cancer [69]. In 2004, Cohen et al. also reported a study where brain metastases were diagnosed in 72 out of 8.225 patients with ovarian cancer (0.9%). The mean age of patients at the time of the metastases diagnosis was 53.7 years and the median interval between diagnosis of ovarian cancer and CNS metastases was 1.84 years. CNS was the only site of metastases in 43% of women, while 65% of patients were found with multiple metastatic sites [10].

Nasu et al. recently published a study where the authors reported that brain metastases from ovarian cancer have a better prognosis than those from corpus or cervical tumors [70]. The average survival time is 8.2–12.5 months for ovarian carcinoma and despite its poor prognosis, a prompt diagnosis of brain metastases can provide an opportunity for appropriate therapy and palliative care [13].

The most common site of metastases in the brain is the cerebral hemisphere with 75% of cases reported in CNS participation in patients with ovarian cancer, followed by 11% and 7.3% in the cerebellum and meninges, respectively [71]. 

Clinical symptoms of CNS metastases include motor weakness, speech disturbance, seizures, headaches, and confusion, and are confirmed in approximately 90% of patients at diagnosis. Nonetheless, some patients can be diagnosed without any neurological deficits [9,11,72,73]. Surgery, irradiation, and chemotherapy represent the treatment of choice for prolonging survival of these patients [12]. Many chemotherapeutic drugs are unable to permeate the blood–brain barrier (BBB), and thus cannot protect the patients from developing brain metastases, limiting the effectiveness for routine use of systemic chemotherapy. Thus, especially for patients with multiple brain metastases, whole-brain radiotherapy (+/− chemotherapy) remains the treatment of choice. However, recent studies of systemic chemotherapy for CNS metastases have shown objective responses and better survival rates among patients with primary germ cell and epithelial ovarian cancer [9,71,72,74].

Several authors have suggested the addition of chemotherapeutic agents, such as carboplatin, cisplatin plus gemcitabine and carboplatin plus docetaxel, to local therapy, in order to improve the local and systemic control of the disease’s spread, since they do not induce further brain damage, such as dementia or brain atrophy [67,75]. Long-term treatment with topotecan, which is a topoisomerase I inhibitor, has been proven beneficial in brain metastases from ovarian cancer with antitumor activity. Due to its ability of freely cross the BBB which can lead to measurable levels of topotecan and its metabolites in cerebrospinal fluid, and its radio-sensitizing capacities, as well as its low toxicity, topotecan can be used as a second-line therapy, even in combination with radiotherapy, with promising results, as the survival can reach up to 26 months [75,76].

#### 3.2.2. Eye

Even though ovarian and peritoneal metastases via lymphatic vessels and hematogenous spread to the brain are commonly observed, ocular metastases are rarely reported. Until 2001, only eight cases of metastatic ovarian carcinoma involving the eye had been published, and only one of them was related to mucinous ovarian cystadenocarcinomas [14]. 

The incidence of ocular metastases from primary cancers varies from 0.07%–2.3% in patients with advanced generalized malignancy to 10%–12.6% in studies of patients with disseminated cancer, while, up to 2009 only nine cases of eye metastasis from primary ovarian cancer were reported [15,16,17,18]. Even though choroidal involvement from gynecologic malignancies is extremely rare, the choroid is the most common site in the ocular metastases, with uveal metastases being 88%, while 9% are located in the iris and only 2% in the ciliary body. The tumors present most often in the posterior pole of the eye with an average of two focuses per eye [14]. In 2009, a review of the literature found only seven reported cases of choroidal involvement from ovarian cancer, four reported cases from cervical tumors and only one case of endometrial adenocarcinoma. All the patients from reported cases presented with common symptoms concerning visual complaints, without a known primary site [18].

The ocular metastases are usually diagnosed via fundoscopy, ultrasound of the eye, CT and MRI scan of the head and orbit. Fluorescein angiography may be helpful showing early hyperfluorescence and diffuse late staining of the lesion. Current literature reports the median survival for prognosis in cases of ocular metastases of all types between 6.5 and 16 months [19,20,21]. Treatment options reveal promising results. Local radiotherapy is an effective approach reducing symptoms, whereas systemic chemotherapy and local radiotherapy appear to be another option. Transpupillary thermotherapy (TTT) with diode laser can also be used. Response to treatment has been reported to be about 81% with remission of ocular disease in 59%. In cases of a painful blind eye, enucleation may be considered [18].

#### 3.2.3. Cutaneous and Subcutaneous Sites

Series investigating the skin metastases consist of a small number of patients due to the rarity of the disease. Cutaneous metastases occur mostly in breast, lung, or kidney cancer [77,78].

Most common sites are the abdominal wall and specifically Sister Mary Joseph Nodule (SMJN) or umbilicus and the skin of chest and breast [22]. It must be noted that when SMJN involvement is found, ovarian malignancy should be considered since it is the origin of 45% of them [79]. Nonetheless, there are case studies reporting metastatic sites on shoulder from ovarian cystadenocarcinoma and in the right occipital region of the scalp deriving from serous carcinoma [23]. In 2016, a cutaneous nasal nodule metastasis was reported by Antonio et al., which after biopsy was identified as metastatic adenocarcinoma. This finding was the cause of discovering an, unknown until then, ovarian malignancy [80]. The clinical presentation of skin metastases varies, and it can present as isolated or multiple cutaneous nodules, inflammatory metastases, but also cicatricial plaques covering the whole surface of the abdomen [81,82].

The exact mechanism of skin metastases has not been fully explained, but several theories exist, suggesting either direct invasion from underlying growth of the tumor, or through the lymphatic pathway from adjacent to extension tumor cells [23]. Another mechanism of metastases well described is the iatrogenic one. Most frequently, minimal invasive surgery is accused of accidental implantation of tumor cells during surgical procedures. The incidence of metastases can reach up to 1.2%, depending on tumor’s aggressiveness. Besides the contamination from port catheters and tissue manipulation, pneumoperitoneum, hypercapnia and CO_2_ acidemia have been shown to participate in the metastases, while no preventive measure (e.g., oxaliplatin) has been found successful enough. Similarly, the use of paracentesis ascites catheters, and fine-needle biopsy have caused the same complications [24,25,26].

Prognosis is considered to be extremely poor, since these types of metastases often occur late in the course of advanced-stage disease, and even worse in nonumbilical metastases, as they often occur as a recurrent disease [27,79]. In a review of 255 patients, Dauplat et al. showed that the interval time of skin metastases from the diagnosis of ovarian cancer—the most important prognostic factor concerning survival—is 32 months (4–77 months), and that median survival time after skin metastases is 12 months (1–41 months) [22]. In another report of nine patients with skin metastases among 220 patients with epithelial ovarian cancer, the overall survival time was four months (2–65 months) [27]. In umbilical metastases, data have shown an increase of survival (mean, 9.7 months) in patients diagnosed with this metastasis before receiving any definitive treatment for the primary tumor, compared to those with a recurrence [23].

There are no treatment protocols for ovarian cancer with skin spread or isolated recurrences. Skin metastases develop late in the diseases’ course, usually after the administration of chemotherapeutic regimen, which renders them rather resistant to any further cytotoxic drugs. Thus, control of the abdominal masses remains a major concern. In focal cutaneous alterations surgical resection should be considered. Cormio et al. reported better survival outcomes when excision was used instead of chemotherapy solely [27]. On the contrary, when extensive skin metastases are present, palliative therapy seems more appropriate since chemotherapy and radiotherapy have a low response rate. In these cases, external beam radiation or hematoporphyrin derivate injection with infrared phototherapy have been described [81]. In the past, electrocoagulation has been successfully used for local control of complications like hemorrhage, pain, and infection of metastatic plaques. However recent data suggest that chemotherapy regimen with taxane and bevacizumab could offer improved survival rates [23,27,78,83].

#### 3.2.4. Bones

Bone involvement in ovarian cancer is rarely reported in the international literature. Even though ovarian tumors are able to give distant metastases, these malignancies have proven to usually spread to bone either directly, transperitoneally, hematogenously, or via lymphatic spread [84]. The vertebral venous system seems to have major importance in the spreading of the ovarian tumors to the bones with some studies reporting the pelvis and vertebral column to be the main target in bone metastases of primary ovarian tumors [28]. Deng et al. recently published a study, where the authors analyzed 1481 patients to find the association between metastatic patterns and overall survival in ovarian carcinomatosis. The most common sites were liver, followed by distant lymph nodes, lung, bone and brain. Even though the authors reported a low frequency of bone participation in ovarian cancer metastases (3.74%), literature research shows that bone participation is even lower with a median survival of 7.5 months [28,29]. Baize et al. presented a patient with left iliac ramus metastases of primary ovarian cancer, while Tiwari et al. reported a case of bone metastases in lumbar spine [85]. Abdul-Karim et al. presented a study of 305 patients with ovarian cancer where bone metastases occurred in seven patients (2.29%) with thoracic vertebra, clavicle, and axial skeleton being the reported metastatic sites [86]. In a study by Zhang et al., 26 patients were diagnosed with metastases from ovarian cancer, with a 0.82% incidence of appearance of bone participation. Twelve cases of cervical vertebra participation were reported, ten in the lumbar vertebra, eight in the pelvis, seven in the thoracic vertebra, five in the limbs, one in the ribs, and two in the sternum confirming the primary hypothesis of the key role of the vertebral venous system in ovarian cancer metastases in bones [28]. Spine and pelvis presented a higher incidence of bone metastases than other sites. Survival after bone metastases of primary ovarian cancer is 33%, 15%, and 8% for one, three and five years, respectively [87,88]. In another study, of 90 cases with ovarian cancer, only one case of bone metastases was detected, and the incidence of bone metastases was 1.1%. In the current English literature, most cases of bone metastases from ovarian cancer were published as case reports and few systematic clinical observations have been performed. Dauplat et al. reported that bone metastases occurred in 1.6% from primary ovarian cancer, confirming the rarity of bone participation in ovarian cancer [22,84]. 

Bone metastases are connected to serious complications, such as fracture and spinal cord compression, and treatment with surgery or radiation therapy is most of the times inevitable. Bone-targeted treatments reduce bone resorption, decreasing the risk of skeletal complications from primary solid tumors and prevent treatment-induced bone loss in patients with bone tumors [89,90]. Denosumab is a human monoclonal antibody that inhibits the nuclear factor-kappaB (RANK) ligand, which is essential for osteoclast differentiation, activity and survival, reducing bone loss in malignant diseases. Denosumab shows a delay in the time of first skeletal-related event while reducing the incidence of radiation to the bone event in comparison to bisphosphonates, being an ideal ally for bone cancer treatment [91].

#### 3.2.5. Rare Lymph Nodes

In 1985, the International Federation of Gynecology and Obstetrics (FIGO) included lymph nodes in the definition of Stage III ovarian cancer [92]. As mentioned before, ovarian tumors spread mainly via direct extension and abdominal implantation, followed by lymphatic spread (mainly through the intra-peritoneal route) and thus, the advanced ovarian tumors are confined in the abdominal/pelvic cavity [57,93]. Distant metastases are observed mostly in patients with advanced stages (IV) of the disease, mainly spread hematogenous or through the lymphatic vessels, and thus, even though it is rarely reported, ovarian cancer is able to metastasize to distant lymph nodes, even in the primary presence of ovarian cancer [94]. Only a few cases of supra-clavicular, axillary, mediastinal and inguinal lymph node metastases have been reported.

●  *Supraclavicular Lymphadenopathy*

To date, only a few cases of supraclavicular lymphadenopathy due to primary ovarian cancer have been reported in the international literature [95,96]. Patel et al. reported, in 1999, five patients with supradiaphragmatic spread from epithelial ovarian cancer. In another study of 100 autopsies of ovarian cancer patients, Dvoretsky et al. reported supraclavicular metastases in only 4% while the total lymph nodes metastases were present in 70% of the patients examined [30,31,32]. Subperitoneal, infradiaphragmatic, and diaphragmatic lymphatic vessels are all connected and thus, the lymphatic fluid route explains supradiaphragmatic metastatic lymph nodes in ovarian cancer [31,97]. Pelvic and para-aortic lymph nodes are present in approximately 40–70% of epithelial ovarian cancers. The left supraclavicular lymph node (LSCLN), the Virchow’s node, collects lymph fluid of the thoracic duct which percolates many organs of the abdomen. [98]. Thus, cancer cells are able to reach and metastasize through lymphatic vessels there.

●  *Inguinal Lymph Nodes*

Ovarian cancer presenting with isolated secluded inguinal lymph node metastases is also rarely reported (0.8–3%) presenting good prognosis after complete cytoreduction followed by chemotherapy [33,34,35,36,37]. Due to the lymphatic drainage mentioned, lymphatic dissemination is a common route of spread in ovarian tumors with lymphatic drainage occurring mainly via the infundibulopelvic ligament to the paraaortic lymph nodes, and thus, these nodes are easily involved with metastases through ovarian vessels. Scholz et al. published a study of a patient that due to a primary undifferentiated serous adenocarcinoma of both ovaries, metastasized to the left inguinal lymph node (and fallopian tube) diagnosed intraoperatively, with the only symptom being left inguinal swelling. Similarly, McGonigle et al. presented a postmenopausal patient who initially presented with an enlarged left inguinal lymph node that was primarily suspected to be metastases from secondary involvement of endometrioid cancer which involved the ovaries and the right salpinx. Kehoe et al. also presented a 66-year-old patient with ovarian cancer that metastasized to the left inguinal lymph node 33 months after diagnosis [37]. 

Musumeci et al. examined 349 patients with ovarian cancer identifying lymph node metastases in 25% of them. Only three of these patients (0.8% of the total participants) had inguinal lymph node involvement that was histopathological confirmed [99]. Zaren et al. published a retrospective analysis of 2232 patients with ovarian cancer and inguinal lymph node metastases. The patients presented inguinal lymph node mass in approximately 5% [100]. Low incidence rate has been similarly reported in autopsy studies. Dvoretsky et al. also reported inguinal lymph node involvement in 3% of cases in his study, but none of these studies provides information on whether inguinal lymph node was an isolated metastases or not [32]. 

●  *Mediastinal—Cardiophrenic Lymphadenopathy*

Metastases of ovarian carcinoma to intrathoracic lymph nodes is uncommon with few studies reporting a low rate of only 2.3% (6 of 255 cases) [38]. Thoracic metastases mainly occur via locoregional spread, usually late in the course of the disease, while, isolated manifestations of distant primary gynecologic malignancies are rare and commonly connected with progressive or recurrent disease, thus, in patients with ovarian cancer, mediastinal involvement is mainly associated with advanced stages and poor prognosis. Metastases to the thorax tend to follow a predictable or characteristic pattern related to the primary ovarian cancer type. Patel et al. reported a study of women with isolated supra diaphragmatic metastases from papillary serous ovarian cancer years after remission of the primary disease [30]. Cardiophrenic lymph nodes (CPLNs), also referred to as paracardiac and supradiaphragmatic lymph nodes, are located above the diaphragm and drain areas from the diaphragm, liver, pleura, and anterior abdominal wall while they empty into the internal mammary chain. The natural history of ovarian cancer includes extensive tumor dissemination on the peritoneal and pleural surface, with possible intrathoracic lymph nodes metastases [101]. CPLN involvement is an important predictor indicator for overall survival and is associated with extensive intra-abdominal tumor spread in the upper abdomen [102]. Treatment of patients with CPLN involvement includes optimal debulking and complete gross resection options. Optimal debulking surgery is associated with a 68.9 months median overall survival (OS) while patients who obtained a complete gross resection (CGR) had a median OS of 72.3 months [39].

#### 3.2.6. Breast and Axillary Lymph Nodes

Although breast cancer is the most common primary malignancy in women, around the world, metastatic tumors to the breast are infrequent, accounting for 0.5 to 1.3% of all breast cancers [103]. The most common extramammary malignancies known to metastasize to the breast are lymphoma, melanoma, lung, and ovarian cancer with the latter’s incidence reaching up to 58% [104,105,106]. The incidence of ovarian cancer metastatic to the breast or axillary lymph nodes is extremely low, 0.03–0.6%. The first case was reported in 1907 and only over 110 cases have been described until 2015 [40,41]. The pathophysiological mechanisms of these metastases suggest a hematogenous or lymphatic pattern of spread. The most common histological type metastasizing to the breast is serous ovarian carcinoma, in 72% of all cases, but other subtypes like dysgerminoma, carcinoid tumor, endometrioid carcinoma, and granulosa cell tumors have also been reported. In 2002, Moreira et al. reported the only metastatic disease to breast from ovarian borderline tumor one year after the initial diagnosis [42,107,108].

It is well known that patients with BRCA mutation I and II can manifest ovarian cancer frequently, in 40% and 9% of carriers, respectively [109]. As it is easily understood, breast and ovarian malignancies can co-exist without the ovary being the primary site. Thus, it is of major importance to discriminate metastases to the breast from primary breast cancer accurately due to significant differences in prognosis and treatment [40].

Immunohistochemical markers can be helpful in the distinction. Estrogen and progesterone receptors cannot be used since they are positive in 50% of ovarian cancers. Conversely, Paired-box gene 8 (PAX8) in most cases of ovarian cancer is frequently expressed, while in breast tumors is negative. Similarly, elevated CA125 and Wilms tumor suppressor gene1 (WT1) positivity usually support ovarian differentiation since they are not present in breast cancer. However, it must be noted that WT1 can be negative in ovarian tumors in up to 20% of the cases. Finally, GATA-3 and Gross Cystic Disease Fluid Protein 15 (GCDFP-15) positivity is a common finding in primary breast tumors in contrast to those of the ovary [110]. 

In 2004, Recine et al., in a retrospective case study, reported that the interval time between ovarian cancer diagnosis and metastases to the breast/axillary lymph nodes was 20 months. However, in some very rare cases, breast metastases could precede the diagnosis of the ovary as the primary site. In 2008, Schneuber et al. described the first case in which, a metastasis to the breast was present 56 months before the definite diagnosis of ovarian malignancy in a 72-year-old patient. Surprisingly, the overall survival of the patient was reported to be 85 months, after the initial diagnosis of breast metastases [42,43]. 

Nonetheless, prognosis after the breast involvement is very poor and it is reported to vary from 13 days to 3.5 months, with an average of 16 months [40,42,43]. Therapeutically, ovarian cancer metastatic to the breast should be treated as a systemic disease, with appropriate chemotherapeutic agents. Surgical treatment with mastectomy should be reserved to those unresponsive to chemotherapy, as a palliative treatment [80,84,88]. 

#### 3.2.7. Rare Intra-Abdominal Metastatic Sites

●  *Spleen*

Metastatic carcinoma to the spleen is unusual, however, specific ovarian tumors are known to metastasize via peritoneum, causing visceral spreading—including metastases to other abdominal organs—, and thus, splenic metastases of ovarian carcinoma is usually found in patients at terminal stage. However, a review of the literature reveals that about half of reported splenectomies for ovarian cancer did not reveal parenchymal splenic involvement but the metastases were mainly reported to be capsular [44]. Therefore, solitary splenic metastases due to ovarian carcinoma are extremely rare and believed to have hematogenously spread origin, being present in only 2% to 3% of patients with epithelial ovarian cancer [44,45]. Of the cases reported until now, only a few referred to patients with solitary splenic metastases, with only a small number presenting distinctive features due to serous papillary adenocarcinoma, one case due to angiosarcoma and one case due to carcinosarcoma. In the majority of these cases, splenic metastases were reported to occur postoperatively while in some patients with splenic participation, the metastases presented simultaneously with the radical resection of the primary ovarian tumor [111]. In some patients, splenic metastases can occur years after surgery. In some patients, splenic metastases were only detected in four cases when ovarian cancer was first diagnosed [45,111]. Cytoreduction is proposed to improve the prognosis in platinum-sensitive patients with solitary metastases to distant organs, including the spleen [45].

●  *Rare Gastrointestinal Metastases*

Frequently, ovarian cancers metastasize to peritoneal surfaces through the abdomen and pelvis. Gastrointestinal invasion is most often superficial, while the mesentery is shortened, and intestinal lumens are narrowed and when invaded, it is usually limited to seromuscular layer of the small and large bowel, with the transmural invasion being less common. 

The dissemination of ovarian carcinoma to the gastrointestinal tract—even though rarely reported—is proved to be achieved through different ways. Primarily ovarian tumor disseminates within the peritoneal cavity, and the gastrointestinal tract is only superficially invaded. Then, the tumor can gradually invade adjacent structures, an invasion that can also be achieved by the lymphatic route. The final way of invasion is via hematogenous dissemination. The average survival after gastric metastases is less than 15 months [46,47]. Ovarian cancers can also rarely metastasize to the parenchyma of the stomach. Kim et al. reported a case of a 58 years old patient with metastatic gastric cancer from an ovarian adenocarcinoma that was diagnosed via endoscopic examination. The mass presented as a submucosal tumor without ulceration and gastrectomy was performed that confirmed the diagnosis of metastatic ovarian serous adenocarcinoma [112]. Small bowel tumors can be primary or metastatic and the distinction is usually difficult. In 2011, the first case of solitary metastases of a clear cell ovarian adenocarcinoma to the small bowel mucosa was reported [113]. Deng et al. published in 2019 a report of a patient who eight months after a total hysterectomy, bilateral adnexectomy and a major mass removal in the pelvic cavity due to epithelial ovarian cancer, presented with small intestinal metastases that were diagnosed via immunohistochemistry [114].

#### 3.2.8. Bronchus and Trachea

The rate of tracheal metastases in cancer patients from all sites is reported to be particularly low (0.6%), as is the incidence of endobronchial metastases of non-pulmonary cancers, ranging between 2% and 5%. Additionally, although the metastatic spread of ovarian tumors to the lungs is common, mostly in late stages, with thoracic involvement incidence reaching up to 50%, the ones in central airways are extremely rare [48,49,115,116]. In 1999, Petru et al. reported the first case of symptomatic intratracheal metastases of ovarian cancer. Ten years later, a rare case of endobronchial metastases by primary papillary serous carcinoma of the peritoneum was described for the first time by Nakao et al. Until 2018, only 10 reports of tracheobronchial metastases from ovarian carcinoma have been made [5,48,49,115].

In a case study by Harrington et al., the pattern of spread to the tracheobronchial tree was reported to be through direct invasion from mediastinal lymph nodes, but hematogenous and lymphogenous route or through the adjacent parenchyma cannot be excluded [115,117].

Clinical manifestations of tracheobronchial metastases are usually dyspnea, cough and hemoptysis but dysphagia and hoarseness can also occur when mediastinal mass is present. Diagnostically, Chest x-rays, CT scan, but mainly bronchoscopy, which can reveal an obstructing tumor, can be useful. Nonetheless, diagnostic problems can occur. Due to the fact that ovarian cancers often form psammoma bodies, invasion in central airways can mimic broncholithiasis. Interestingly, in 2018 an extremely rare case of metastatic ovarian malignancy to bronchial tree complicated by aspergillosis was reported [38,48,118].

The time interval between diagnosis of primary cancer and endobronchial metastases is reported to be 11 months–26 years, and overall survival after diagnosis to be 6–24 months [48,49].

The therapeutic approach has been heterogenous between different cases. Some patients have been treated with surgery (pneumonectomy), and others with combination of chemotherapy and radiation. Finally, it is worth mentioning that in 2007, Choi et al. reported the first successful endoablative removal of an endotracheal metastasis of ovarian cancer with bronchoscopic electrocautery [48,49,115,117].

#### 3.2.9. Heart and Pericardium 

Primary cardiac tumors are considered a rare event (0.001%–0.28%), but secondary ones’ incidence ranges between 2.3% and 18.3%. The most common type of cardiac metastases is pericardial involvement, in 70% of all cases, followed by myocardial (32%) and epicardial (5%) metastases. The prevalence of ovarian cancer as primary site, among all neoplasms metastatic to the heart, is reported to be 10.3%, with the most common clinical manifestation being pericardial effusion resulting possibly in cardiac tamponade, as is easily extracted from relevant literature [50,51,119]. In two autopsy series with 255 and 100 patients, the incidence of pericardium as a distant metastasis, among ovarian cancer patients, was 2.4% and 4% respectively. The interval time between diagnosis of ovarian tumor and distant metastases was shown to be 41.4 months (6–77 months), while the overall survival after pericardial metastases, reached 2.3 months (1–5 months) [22,32]. 

In a case report from 1978, Griffith et al. described what it is considered to be the first case of right ventricular outflow tract obstruction by a metastatic deposit from ovarian carcinoma. It was only after the patient’s death, that diagnosis was certified, suggesting an ovarian primary with a solitary metastasis to the myocardium [120].

Nonetheless, pericardium remains the most common cardiac metastatic site of ovarian cancer. Although international literature is scant, pericardial effusion and cardiac tamponade are most commonly presented as the dominant symptoms in many studies. It must be noted that in the previously mentioned autopsy, all patients with pericardial involvement had also presented pleural effusion [22,50,121,122,123].

Pericardial effusion frequently leads to cardiac tamponade, in which a large amount of fluid (200 mL or more) is accumulated in a short period of time in the pericardium. Dyspnea, orthopnea, fatigue or severe chest pain can be some of the clinical symptoms. Primary diagnostic measures are physical examination, echocardiography and electrocardiography [51,122,123].

The treatment of this life-threatening event is based on pericardiocentesis, preventing re-accumulation of fluid through prolonged drainage, sclerotherapy, or formation of pericardial window [121,122]. 

Prognosis and survival rates in ovarian malignancies with pericardial effusion are reported to be poor. Little is known about the ideal therapy, because of the rarity of the disease. However, as presented by Perri et al., long-term survival was achieved in patients who received chemotherapy after resolution of the pericardial effusion, with overall survival of 3–72 months, after cardiac tamponade was treated [50,51]. 

#### 3.2.10. Placenta and Fetus

The incidence of cancer in pregnancy is estimated around 1/1000 pregnancies, with most common malignant tumors being breast cancer, cervical cancer, hematological malignancies, and melanoma [124]. Ovarian cancer’s frequency in pregnancy is assessed between 1/80,000 and 1/25,000 gestations, and the most common histological types are dysgerminoma, granulosa cell, and borderline ovarian tumors [52].

During the last decade, around 100 cases of metastatic involvement of the products of the conception have been reported. Metastases of maternal malignancies to placenta or fetus occur through bloodstream [53,54,124]. 

To our best knowledge, gestational ovarian cancer with placental involvement has only been reported three times in the international literature, without any case of fetal metastases [52,53,54]. The reason of this extremely low incidence is not completely clarified, but it may be related to the immunosuppressing responses of the pregnancy, as well as due to the protective mechanisms of placento-fetal unit which render the placenta an effective barrier, preventing cancer spread to the fetus [52,54,124].

In 1960, Horner published the first case report of a maternal ovarian tumor with placental metastases without any fetal involvement. In a 28-year-old woman, at the 32nd week of gestation, the membranes ruptured spontaneously, and cesarean section was performed. The abdominal exploration revealed a semisolid cystic ovarian tumor, histologic report of which showed infiltration of a papillary cystadenocarcinoma well differentiated. Placenta’s microscopic analysis revealed large masses of tumor cells in the intervillous villi [53]. 

The second case was reported in 1989 by Patsner et al. A 28-year-old woman, at 24th weeks of gestation underwent an abdominal exploratory operation due to a large pelvic mass deriving from left ovary, and a poorly differentiated adenocarcinoma was diagnosed. At the 25th week of gestation, the patient displayed signs of chorioamnionitis and sepsis, thus prostaglandin termination was initiated, resulting in the stillbirth of the infant. Microscopic examination of the placenta showed focal metastatic adenocarcinoma while fetal analysis showed no anomalies, congenital or tumor [52]. 

The last known case report was made in 2017, by Honda et al., concerning the first recurrence of ovarian cancer with placental metastases ever reported. A 39-year-old woman with a history of ovarian adenocarcinoma, at the 36th week of gestation, underwent cesarean delivery due to premature membrane rupture and non-reassuring fetal status. A 15 cm abdominal mass was discovered, histologic analysis of which detected a poorly differentiated ovarian adenocarcinoma, similar to the patient’s primary one. Placental examination revealed a cluster of cancer cells, similar to that the ovary, without invasion of the chorionic villi. The patient refused adjuvant chemotherapy and passed away after three months. No signs of metastases were found in the infant at the time of birth and up to the age of six years [54].

Due to the rarity of such cases, the report in the literature of any future case of metastatic disease to the placenta or to the fetus should be warranted. 

## 4. Conclusions

Ovarian cancer is known to metastasize throughout the peritoneum, to the abdominal and pelvic organs. Nevertheless, the organ distribution from the primary site is not completely random. The most common routes of spread are proven to be a direct invasion, however, lymphatic and hematogenous vessels also play a major role in the metastatic pathway, especially for the least common metastatic sites described in this review.

Ovarian cancer is one of the leading causes of death in western countries. The most common sites of distant metastases are pleura, liver, lung and lymph nodes, while eye, skin, breast, bones, CNS, heart, central airways, rare intra-abdominal tissues, placenta, and specific lymph nodes have also proven to be targets of ovarian peritoneal cancer cells. Thus, physicians should always be aware of every symptom and clinic manifestations arising from these organs. 

The occurrence of metastatic sites described in this review is numerically inconsequential opposing to most common intraperitoneal sites which occur via direct invasion from peritoneal dissemination. Even though most of the rare distant metastatic sites are not fully exploited due to the rarity of the reports, they represent a challenge for the gynecologists-oncologists in the clinical practice. Furthermore, their study offers an opportunity for significant biological research considering the behavior of ovarian and peritoneal cancers’ pathophysiological mechanisms.

## Figures and Tables

**Figure 1 cancers-11-01044-f001:**
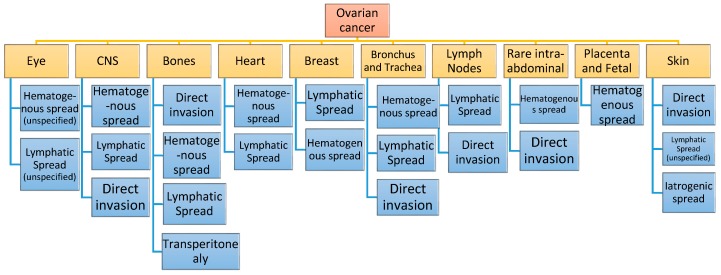
Rare distant metastatic sites of ovarian cancer and their route of metastasis.

**Table 1 cancers-11-01044-t001:** Metastatic sites, frequency and prognosis and peritoneal carcinomatosis.

Metastatic Site		Frequency	Prognosis (Median Survival)	Ref.
CNS		0.3–12%	8.2 months	[9,10,11,12,13]
Eye		9 cases reported until 2009	6.5–16 months	[14,15,16,17,18,19,20,21]
Skin		1.2%	12 months (1–41 months)	[22,23,24,25,26,27]
Bones		<3.74%	7.5 months	[28,29]
Lymph nodes	Supraclavicular	4%	NR	[30,31,32,33,34,35,36,37,38,39]
	Inguinal	0.8–3%	NR	
	Mediastinal-Cardiophrenic	2.3%	68.9–72.3 months	
Breast		0.03–0.6%	16 months (13 days–3.5 months)	[40,41,42,43]
Rare intra-abdominal	Spleen	2%–3% in epithelial ovarian cancer	NR	[44,45,46,47]
	Gastrointestinal	Depends on type and stage of diagnosis	<15 months	
Bronchus and Trachea		10 reports until 2018	6–24 months	[5,48,49]
Heart		2.4–4%	3–72 months	[22,32,50,51]
Placenta and Fetus		3 cases reported until now to placenta 0 to fetus	NR	[52,53,54]

CNS: Central Nervous System, NR: Not Reported.

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
