# Peer review of "Rare Distant Metastatic Disease of Ovarian and Peritoneal Carcinomatosis: A Review of the Literature"

_cancers, 2019, doi:10.3390/cancers11081044_

Round 1

Reviewer 1 Report

Review Cancers

As such the submitted manuscript covers a very interesting issue. However, I have several serious comments and points of criticism regarding this review:

Title: there is a major orthographic mistake: “carcinomatocis” in steed of “carcinomatosis”, furthermore, authors should declare that this contribution is “A review of literature” already in the title. In the Abstract: “arouse” in steed of course. The same spelling error in the Introduction line 56.

Table 1: In this table bizarre indications are included (e.g. advanced, generalized, disseminated etc.), which should urgently be omitted. If there is a need for, please indicate them in the text body.

Figure 1: confusing! For example, what should “Abdominal implantation” mean in the context of that figure. In the legend “metastatic pathways” is not really the right expression. I think authors mean the route of metastasis (?).

In the “Results” section (I don’t think that the subhead “Results” is applicable in this review), reference 13 (line 92) are not underscoring the here reported fact. Also references 14 & 15 (line 96) are not adequate.

In the paragraph 3.1. some old fashion and also strange hypothetic models are revitalized, which do not reflect the current knowledge and are overloading the manuscript.

The recital of cases and various reports for each possible site of metastasis, are mostly not of clinical or scientific relevance and do not add new information leading to a better understanding why just this tumor spread to this specific extraordinary side, or of ovarian cancer as a disease in general.

On the other hand important facts such as topotecan and its high permeation of the blood-brain barrier, or the possible treatment of bone metastases with denosumab (or bisphosphonates) are lacking. In addition, data on the cardiophrenic lymphnode spread (which is very important and is of clinical significance) are also completely lacking.

In summary the submitted manuscript is by far too long because of multiple redundancies and should urgently be shortened to a 3-4 pages manuscript.

Such a review of literature should represent a condensation and a sound interpretation of the various quotations and not a simple consecutive recital of cases reported elsewhere.

Author Response

Reviewer 1:

1.       The Manuscript was copyedited appropriately, and minor grammatical errors have been corrected by a native English speaker. The words “carcinomatocis” and “arouse” have been appropriately replaced by the words “carcinomatosis” and “course” respectively.

2.       The title was modified according to the reviewer’s remark to “Rare distant metastatic disease of ovarian and peritoneal carcinomatosis: A review of the literature”.

3.       Table 1 was modified and changes were made in order to be more comprehensive and the unnecessary indication which were mentioned by the reviewer, were removed of the table 1 and added in the text according to the comments.

4.       The legend of figure 1 was appropriately changed according to the reviewer’s comments and minor changes were made in order to avoid further confusion.

5.       The “Results” section was changed to “Pathways and metastatic sites of ovarian and peritoneal cancer”.

6.       Recent references were added inside the text and previous references 13,14,15 were repositioned for the text to be more thorough and substantiated.

7.       The paragraph 3.1 was modified and recent evidence-based citations were added in order to support the reported facts.

8.       The cases reported aim on supporting the evidence of rare distant metastasis of ovarian and peritoneal cancer, which is the main goal of this review. Due to the word limitation, and lack of the relevant literature, it was unable to analyze the main metastatic routes of each reported rare metastatic site.

9.       CNS metastases treatment with Topotecan and its high permeation of the blood-brain barrier has been described according to the reviewer’s suggestion. Moreover, data on bone metastases therapy with Denosumab, and data on the cardiophrenic lymph node spread have been described in the appropriate parts of the text.

10.   Some of the repetitions inside the text aim on a better understanding of the review, and all authors believe that their subtraction could possibly result to reading confusions.  The manuscript was formatted according to journal guidelines (4000-6000 words). The manuscript could not be shortened to a 3-4 pages manuscript due to the large amount of data and references supporting our findings regarding this topic.

Reviewer 2 Report

This manuscript described the rare distant metastatic sites of ovarian cancer that is usually metastasized in the peritoneal cavity, offering valuable information. This paper is well organized and written easy to understand.

Just minor things

Use decimal “.”, not “,” throughout the whole paper including Table.

Author Response

1.       Use of decimals “.” Instead of “,” were replaced throughout the whole paper including the Table.

Reviewer 3 Report

The Authors report on distant metastases in ovarian carcinoma. this topic is very important in order to plan the management of ovarian cancer patients. However there are a number of important topic which need to be included in the paper:

- the distribution of distant metastases should be differentiated between sites at primary diagnosis of ovarian cancer and sites at the time of recurrent diesase.

- CNS involvement should be discussed in detail regarding possible treatment modalities and new different aproaches (see Cormio G Int J Gynecol Cancer 2011)

Author Response

1.       The distribution of distant metastases between sites at primary diagnosis of ovarian cancer and sites at the time of recurrent disease could not be analyzed due to word limitation by the journal guidelines (4000-6000 words). Moreover, the purpose of this review article aims mainly on informing the scientific community about the data of the latest decades regarding the targeted rare distant metastatic site of ovarian cancer, thus, the distinction between primary and recurrent cancer does not address this certain topic of interest. More importantly each rare case reported, considering the metastatic sites, does not necessarily reflect the main routes of ovarian cancer metastases.

2.       CNS involvement was discussed in detail, regarding possible treatment modalities and new different approaches according to the reviewer’s comments. References suggested by the reviewer were added.

Reviewer 4 Report

In this manuscript the authors have attempted to review the literature to acknowledge rare distant metastatic sites of ovarian and peritoneal carcinomatosis. Authors have done a fairly extensive search to include appropriate literature. 

Overall this is a very well written manuscript. The structure and content of the manuscript is good. Few comments are noted below:

It would have been great if authors could have included the molecular/genetic features/drivers of the tumors of rare metastatic sites (characteristics)? That could potentially be valuable towards a therapeutic angle. 

What is the overall percentage of rare metastatic tumors compared to common sites? any known differences in genetic features between metastatic tumors of major and rare sites?

Lines 93 - 96: This section on HGSOC origin need to be re-written and requires elaboration and clarity. Also, more appropriate references are needed. some references (reviews) are provided below:

https://www.ncbi.nlm.nih.gov/pubmed/29061967 

https://www.ncbi.nlm.nih.gov/pubmed/26669862 

Author Response

1.       Molecular and genetic features/drivers are not fully clarified in the cited articles. Furthermore, their mention represents a volume of information that requires a separate and detailed analysis, in a different article-review. In order to respect the journal’s guideline word limitation (4000-6000 words), and avoid going off topic, this subject could not be addressed. Instead, the authors focused on the clinical data of patients with rare-distant metastatic sites of ovarian and peritoneal cancer.

2.       The overall percentage of each rare metastatic site has been reported in Table 1. However, the overall percentage of rare metastatic tumors compared to common sites, could not be calculated due to the rarity and the heterogeneity of each metastatic site mentioned in this review.

3.       Previous lines 93-96 have been modified and re-written according to the reviewer’s suggestion, with more appropriate data added in order to be more clarified and comprehensive. References suggested by the reviewer have been added.